# Vision-Related Quality of Life in Patients with Systemic Lupus Erythematosus

**DOI:** 10.3390/healthcare12050540

**Published:** 2024-02-24

**Authors:** Francisco de Asís Bartol-Puyal, María Chacón González, Borja Arias-Peso, Damián García Navarro, Silvia Méndez-Martínez, María Pilar Ruiz del Tiempo, Luis Sáez Comet, Luis Pablo Júlvez

**Affiliations:** 1Ophthalmology Department, Miguel Servet University Hospital, 50009 Zaragoza, Spain; 2Miguel Servet Ophthalmology Research Group (GIMSO), Aragón Institute for Health Research (IIS Aragón), 50009 Zaragoza, Spain; 3Surgery Department, Universidad de Zaragoza, 50018 Zaragoza, Spain; 4Internal Medicine Department, Miguel Servet University Hospital, 50009 Zaragoza, Spain; 5Biotech Vision SLP, University of Zaragoza, 50009 Zaragoza, Spain

**Keywords:** hydroxychloroquine, impact of visual impairment questionnaire, systemic lupus erythematosus, vision-related quality of life

## Abstract

Purpose: To assess vision-related quality of life (VRQoL) in patients with systemic lupus erythematosus (SLE) under treatment with hydroxychloroquine (HCQ), and to find the influencing factors. Methods: Cross-sectional study enrolling SLE patients for less than ten years (Group 1), SLE patients for more than ten years (Group 2), and healthy controls (Group 3). SLE patients should be under treatment with HCQ but without ophthalmological affection. Schirmer test, best-corrected visual acuity (BCVA), axial length (AL) with optical biometry, and swept-source optical coherence tomography–angiography (OCTA) Triton (Topcon) were performed. All participants fulfilled the Impact of Visual Impairment questionnaire, and SLE patients answered the Lupus Impact Tracker (LIT) questionnaire. Additional data were obtained from clinical records. Results: A totals of 41 eyes (41 patients), 31 eyes (31 patients) and 45 eyes (45 volunteers) were enrolled in the study groups. The mean ages were 41.09 ± 9.56, 45.06 ± 8.47 and 40.25 ± 10.83 years, respectively (*p* = 0.10). The LIT outcomes were 33.49 ± 20.74 and 35.98 ± 22.66 (*p* = 0.63), respectively. Group 3 referred to a better VRQoL than Group 2 in all categories and than Group 1 in some of them. A linear regression analysis showed that serum ferritin, SLE activity scales, body-mass index (BMI), age, and BCVA influenced VRQoL. The LIT questionnaire was correlated to two categories of the Impact of Visual Impairment questionnaire (IVI). Conclusions: Despite no ophthalmological affection, SLE patients refer to poorer VRQoL because of disease activity and a low health-related quality of life, which has a negative influence on VRQoL. This masks the effect of other ophthalmological conditions such as dry eyes. Other variables influencing VRQoL are age and BMI, and BCVA, to a lesser extent.

## 1. Introduction

Systemic lupus erythematosus (SLE) is a disease with multisystemic affections. It usually affects young women, and has a genetic basis. Its prevalence may range between 1.5 and 7.4 cases per 100,000 person-years [1]. Because of a dysregulated immunity, different autoantibodies are typically elevated, such as antinuclear antibodies (ANA), anti-DNA, anti-Smith (anti-Sm), anti-Ro, or anti-La. Kidney affection is the main survival predictor, but patients may also die from infections or other complications. SLE may be also associated with Sjögren syndrome or antiphospholipid syndrome, among others. Consequently, patients with SLE may refer to a reduced health-related quality of life (HRQoL) [2]. Treatment with hydroxychloroquine (HCQ) increases life expectancy in these patients [3,4], as well as increasing their HRQoL [5]. Therefore, it is usually used as a first-line treatment.

Patients with SLE usually refer to a poorer HRQoL in comparison with healthy individuals, especially those with high disease activity [6], fatigue, depression, or other neurological or psychological co-morbidities [7,8,9,10]. However, vision-related quality of life (VRQoL) has not been studied so far, despite SLE may affect eyes as well in different ways: eyelid lupus, dry eye, episcleritis, scleritis, neuro-ophthalmological affection, retinopathy, or choroidopathy, among others [11]. A low VRQoL has been reported in patients with dry eye [12], different retinal diseases [13,14], episcleritis and scleritis [15], or optic neuritis [16]. Hence, a poorer VRQoL should be expected in SLE patients with ophthalmological affections. In addition, HCQ is a potentially toxic agent for the retina, and corticosteroids may accelerate the presence of cataracts. Thus, vision may worsen reversibly or even irreversibly.

Different VRQoL questionnaires have been created so far, such as the 25-item National Eye Institute Visual Function Questionnaire (NEI VFQ-25), the National Eye Institute Visual Function Questionnaire, the Low Vision Quality of Life Questionnaire (LVQOL), the Impact of Vision Impairment (IVI) [17], the Functional Vision Questionnaire for Children and Young People (FVQ_CYP), or the Refractive Status and Vision Profile (RSVP). Every questionnaire is focused on a type of patient or an ophthalmological affection.

The main purpose of this study is to assess VRQoL in patients with SLE under treatment with HCQ. The secondary purpose is to find those factor that may have an influence on it.

## 2. Materials and Methods

A cross-sectional study was performed between November 2020 and January 2022 at a third-level hospital. It adhered to the tenets of the Declaration of Helsinki and it received the approval of the Ethics Review Board (EPA21/013). Patients signed written informed consent after being informed about the study. Patients were recruited from the department of internal medicine of the hospital.

Three study groups were enrolled. The first group comprised patients diagnosed with SLE by the department of internal medicine less than ten years before this study and fulfilling the following inclusion criteria: Caucasian race, treatment with HCQ, age lower than 55 years, and an axial length (AL) between 21 and 25 mm. Diagnostic criteria for SLE were those of EULAR/ACR. Exclusion criteria comprised of the following: any ophthalmological disorder, previous ophthalmological treatment, retinopathy secondary to HCQ, ophthalmological disorders secondary to SLE, any other disease different from SLE or Sjögren syndrome, or treatment with corticosteroids with a dose higher than 5 mg/d. The second study group comprised similar patients but with SLE for more than ten years. The third study group was composed of age-matched healthy volunteers.

Retinopathy secondary to HCQ was considered as an alteration to the visual field, autofluorescence, optical coherence tomography, fundoscopy, or multifocal electroretinogram. 

First, Schirmer type I test was performed, and afterwards patients underwent a deep ophthalmological examination comprising best-corrected visual acuity (BCVA) under photopia, slit-lamp examination, Goldmann tonometry, and indirect fundoscopy. They were also examined with IOLmaster 500 (Carl Zeiss, Jena, Germany) and Deep Range Imaging (DRI) swept-source optical coherence tomography (SS-OCT) Triton (software version 1.1.7, Topcon Corp., Tokyo, Japan) through dilated pupils and under scotopic lighting conditions. These examinations were performed between 16:00 and 20:00 h. A fovea-centred 7 × 7 mm macular cube analysis was performed, and retinal boundaries were delimited automatically by its internal algorithm. Automatic measurements of retinal thicknesses were given in ETDRS sectors. A 6 × 6 mm OCT-angiography (OCTA) was performed as well, and vascular density (VD) values were obtained automatically. Finally, patients were asked to fill out the Impact of Visual Impairment (IVI) questionnaire. In addition, patients with SLE fulfilled the Lupus Impact Tracker (LIT) questionnaire.

The IVI questionnaire is a VRQoL questionnaire composed of 28 different items. Answers are graded in three or four possible steps (never, a little, moderately, a lot, not applicable), and a Rasch analysis converts them to a value ranging from 0 to 100, with 100 being the best VRQoL. Three different categories can be established: reading and accessing information (items 1, 3, 5–9, 14–15), mobility and independence (items 2, 4, 10–13, 16–20), and emotional well-being (items 21–28). This questionnaire does not differ quality of life due to one eye or the other, but by both at the same time.

The LIT questionnaire is composed of 10 items and it can monitor the impact of SLE and its treatment on patients’ HRQoL. It is a reliable tool that was specifically designed for SLE [18]. Nevertheless, this questionnaire has no vision-specific items. Mean scores were converted in a scale ranging from 0 to 100, with 100 being the best possible HRQoL.

Additional data were obtained from clinical records. These data should not have been recorded for longer than two months before the examination: body mass index (BMI), hemoglobin (Hb), hematocrit, iron (Fe), ferritin, vitamin D, calcium (Ca), duration of SLE, duration of treatment with HCQ, daily dose of HCQ, mean dose of HCQ, cumulative dose of HCQ, Systemic Lupus Erythematosus Disease Activity Index (SLEDAI), and Systemic Lupus International Collaborating Clinic index (SLICC).

One eye of each patient was randomly chosen for statistical analysis, and it was performed using the Statistical Package for the Social Sciences (SPSS) software for Windows (software version 20, IBM Corporation, Somers, NY, USA). Statistical significance was established at *p* < 0.05. Means and standard deviations were calculated, and ANOVA tests with Bonferroni post hoc tests were performed for comparisons between groups. Pearson’s correlations were performed, as well as multiple lineal regression analyses.

## 3. Results

No participant was excluded because of the presence of eye disorders. In total, 41 eyes of 41 patients with SLE for less than 10 years (Group 1), 31 eyes of 31 patients for more than 10 years (Group 2), and 45 eyes of 45 healthy volunteers (Group 3) were enrolled in the study. There were 3 men and 38 women, 1 man and 30 women, and 1 man and 40 women, in Groups 1, 2 and 3, respectively. In total, 23 right eyes (OD) and 18 left eyes (OS), 18 OD and 13 OS, and 27 OD and 18 OS were enrolled in Groups 1, 2 and 3, respectively. There were no differences between the groups in terms of demographic or ophthalmological data (Table 1).

Table 2 displays the outcomes in the IVI questionnaire. In general lines, there were no differences regarding VRQoL between patients with fewer or more than ten years of SLE. However, there were significant differences between healthy and patients with SLE for more than ten years in nearly all the items and categories.

Table 3 displays automatic measurements using SS-OCT DRI Triton. There were no differences between the groups.

Table 4 displays some clinical comparisons between both SLE groups.

The factors influencing the IVI questionnaire in Groups 1 and 2, in linear regression analysis, are displayed in Table 5.

The bivariate correlations between the IVI and LIT questionnaires were analyzed within Groups 1 and 2. These are displayed in Table 6. Both were moderately or highly correlated, except for the mobility and independence category. The correlation values were higher in Group 2 than in Group 1.

## 4. Discussion

SLE is a chronic disease with multisystemic affections, and that is why patients may refer to different limitations in their daily activities. HRQoL in patients with SLE has been studied with different questionnaires, and even specific questionnaires have been developed for SLE [18,19]. In is sometimes significantly impaired, even in the state of remission, and these cases are not sufficiently considered in current SLE therapeutic guidelines [17]. Nevertheless, most HRQoL questionnaires minimize the impact of altered vision on global health; that is, they focus on the impact of the disease on general daily activities, but they do not specifically ask if this impact is due to poor vision. Ophthalmological disorders may occur in these patients either as part of the multisystemic involvement of SLE, or as a secondary effect of the treatment with HCQ.

The IVI questionnaire is a high-quality instrument [20] for measuring VRQoL, which has been validated in different countries and languages [21,22]. Its psychometric properties are adequate [23] and it shows good internal consistency [24]. It has been tested in different ophthalmological pathologies typically known for decreased vision such as age-related macular degeneration [25], retinal detachment, glaucoma, myopic macular degeneration [26], or cataracts [27]. Our outcomes in this questionnaire show that VRQoL decreases even prior to objective ophthalmological affection. The three categories and nine of the twenty-eight items that comprise the questionnaire show worse outcomes in patients with SLE for more than ten years than in healthy subjects. Furthermore, patients with SLE for less than ten years refer to poorer VRQoL than healthy individuals for reading and accessing information; that is, one category and two items. In the general lines, there were no differences within patients with SLE (Groups 1 and 2). The exception was the worry of their vision worsening. It seems that the presence of a chronic disease with multisystemic involvement such as SLE does not only imply a worse HRQoL, but also a poorer VRQoL despite not presenting an ophthalmological affection. The more time the patient suffers from the disease, the poorer the VRQoL becomes. As far as we know, this is the first time that VRQoL has been studied in SLE patients with no ophthalmological involvement.

Among factors influencing VRQoL, serum ferritin was the most important in patients with SLE for less than ten years. It had a negative impact, and even patients in our cohort did not present with anemia or altered levels of serum ferritin, nor were there differences between both groups. Serum ferritin is considered a marker of disease activity of different autoimmune diseases including SLE [28], and it correlates with SLEDAI scores [29,30]. High ferritin levels may reflect an increase in disease activity, and eventually may lead to a poorer HRQoL. Additionally, systemic disease activity scales (SLEDAI and SLICC) showed negative influences on VRQoL in patients with SLE for more than ten years in our study.

It has been found that a high BMI decreases HRQoL in patients with SLE [31]. Obesity may cause fatigue, a worsening of the complications of SLE, and the diminution of other physical aspects. Our study has found that a high BMI has a negative impact on VRQoL, too. We could not differentiate between patients with normal weights and obesity because none of them suffered from obesity.

Serum vitamin D levels were low in both SLE groups, and this showed some influence on VRQoL to a certain extent. This could be related to the fact that patients under treatment with HCQ, corticosteroids, or immunosuppressants exhibit lower serum vitamin D levels [32]. A higher disease activity requires higher doses of medical treatment, and consequently serum levels of vitamin D may drop. In addition, vitamin D deficiency is associated with a high SLE-activity disease [33].

OCT or OCTA parameters had very little influence on VRQoL. AL, IOP, and Schirmer tests showed no influence either. Nevertheless, we found some factors influencing VRQoL, such as BCVA and age, as expected. The Schirmer test should appear among these expected variables [34]. A possible explanation would be that systemic disease activity plays a major role and consequently minimizes the impact of dry eye on VRQoL in patients with SLE.

Retinal thickness outcomes in our study were within the ranges of normality, as well as VD outcomes [35]. Retinal thickness may become thinner as a result of neurodegeneration [36], and VD may decrease due to inflammatory activity [37] and HCQ treatment [38]. However, we found no differences among the three study groups. This may be explained by a good disease activity control that would prevent patients from suffering long-standing consequences.

The LIT questionnaire was specifically designed for measuring HRQoL in SLE patients. Although it was not particularly designed for measuring VRQoL, we found moderate and high correlations between both questionnaires in the categories of reading and accessing information and emotional well-being. Therefore, we believe that the impact that SLE has on HRQoL implies an associated impact on VRQoL, despite no ophthalmological affections.

The strengths of our study are that all patients were receiving the same type of treatment, and that the inclusion and exclusion criteria were restrictive in order to obtain a uniform study sample. To the best of our knowledge, this is the first time that VRQoL has been studied and related to SLE activity disease using a vision questionnaire.

The main limitation of our study is that it is not possible to differentiate the effects of SLE and that of HCQ in our results. Furthermore, we did not include patients treated with immunosuppressive drugs or high levels of corticosteroids, which are used in the case of poor control of the disease activity. In our study, patients with SLE for more than ten years had been under treatment with HCQ for a relatively short period; possibly because they maintained a good disease control.

Future research should confirm these findings in other autoimmune disorders. It would be very useful to develop another HRQoL questionnaire including VRQoL aspects so that it would reflect more precisely the global QoL in these patients.

## 5. Conclusions

VRQoL is decreased in patients suffering from chronic diseases with multisystemic affections such as SLE, due to systemic disease activity and a poorer HRQoL, in spite of not presenting with ophthalmological involvement. A low HRQoL has a high influence on patients’ perceptions of their vision. This even masks the effect of other ophthalmological conditions such as dry eye. Other variables influencing VRQoL are age and BMI, as well as BCVA, to a lesser extent.

## Figures and Tables

**Table 1 healthcare-12-00540-t001:** Demographic and ophthalmological data.

	SLE Patients		Healthy Volunteers	*p*
	<10 Years	>10 Years		
Age, years old	41.09 ± 9.56	45.06 ± 8.47	40.25 ± 10.83	0.10
BMI, kg/m^2^	25.94 ± 5.64	26.21 ± 6.98	23.15 ± 3.41	0.05
BCVA, logMAR	0.01 ± 0.12	0.01 ± 0.15	−0.04 ± 0.09	0.19
IOP, mmHg	13.12 ± 2.48	14.29 ± 2.76	13.96 ± 3.02	0.18
AL, mm	23.55 ± 1.00	23.28 ± 0.91	23.80 ± 0.96	0.07
Schirmer I, mm	7.73 ± 7.25	9.81 ± 7.67	10.67 ± 6.77	0.16

SLE: Systemic Lupus Erythematosus, BMI: body-mass index, BCVA: best-corrected visual acuity, IOP: intraocular pressure, AL: axial length.

**Table 2 healthcare-12-00540-t002:** IVI questionnaire outcomes.

	SLE Patients		Healthy Volunteers	*p* (ANOVA)	Post hoc (Groups)		
	<10 Years	>10 Years			1–2	1–3	2–3
Reading and accessing information	86.47 ± 15.09	82.20 ± 19.39	94.34 ± 8.91	**<0.01**	0.65	**0.04**	**<0.01**
Mobility and independence	92.22 ± 13.42	89.45 ± 19.16	98.25 ± 5.01	**0.01**	1.00	0.10	**0.01**
Emotional well-being	89.57 ± 10.91	83.27 ± 17.32	95.03 ± 8.11	**<0.01**	0.09	0.12	**<0.01**
Item 1. Watching and enjoying TV	83.59 ± 24.26	79.61 ± 29.85	94.08 ± 14.23	**0.02**	1.00	0.11	**0.02**
Item 2. Recreational activities	83.36 ± 27.77	90.76 ± 23.42	97.12 ± 11.19	**0.01**	0.45	**0.01**	0.62
Item 3. Shopping	88.74 ± 23.42	84.31 ± 28.67	95.64 ± 12.75	0.08			
Item 4. Visiting Friends or family	95.29 ± 18.47	95.82 ± 13.34	98.74 ± 8.46	0.47			
Item 5. Recognizing or meeting people	86.46 ± 26.79	86.09 ± 27.84	93.63 ± 15.42	0.26			
Item 6. Looking after appearance	90.57 ± 21.59	89.81 ± 23.51	95.65 ± 12.75	0.34			
Item 7. Opening packaging	97.80 ± 10.10	92.72 ± 21.51	98.43 ± 7.34	0.16			
Item 8. Reading medical labels	73.11 ± 36.21	65.72 ± 37.45	85.96 ± 26.43	**0.03**	1.00	0.23	**0.03**
Item 9. Operating housework	91.18 ± 18.67	88.67 ± 23.85	96.77 ± 12.48	0.14			
Item 10. Interfered with getting outdoors	93.90 ± 20.16	95.16 ± 15.33	97.12 ± 11.19	0.64			
Item 11. Avoid falling or tripping	92.62 ± 20.19	90.24 ± 22.79	98.38 ± 7.59	0.11			
Item 12. Travelling or using transport	97.33 ± 9.61	95.16 ± 15.33	97.12 ± 11.19	0.71			
Item 13. Going down steps, stairs, or curbs	89.46 ± 22.28	84.23 ± 25.77	99.19 ± 5.43	**<0.01**	0.75	0.06	**<0.01**
Item 14. Reading ordinary-sized print	77.49 ± 28.93	74.92 ± 30.24	91.07 ± 20.31	**0.01**	1.00	0.06	**0.03**
Item 15. Getting information	89.20 ± 19.58	79.79 ± 33.13	97.95 ± 9.76	**<0.01**	0.21	0.19	**<0.01**
Item 16. Safety at home	93.34 ± 22.83	87.97 ± 30.64	99.19 ± 5.43	0.07			
Item 17. Spilling or breaking things	91.96 ± 24.10	83.43 ± 35.56	99.19 ± 5.43	**0.02**	0.39	0.47	**0.01**
Item 18. Safety outside the home	91.36 ± 23.92	81.74 ± 36.06	97.93 ± 9.95	**0.02**	0.28	0.61	**0.01**
Item 19. Stopped doing things	92.29 ± 17.71	88.33 ± 28.58	95.57 ± 9.19	0.11			
Item 20. Needed help from other people	93.51 ± 9.67	90.76 ± 23.42	99.19 ± 5.43	0.09			
Item 21. Felt embarrassed	96.63 ± 12.41	96.81 ± 12.61	99.13 ± 5.84	0.48			
Item 22. Felt frustrated or annoyed	94.72 ± 14.62	87.27 ± 24.32	95.65 ± 12.45	0.09			
Item 23. Felt lonely or isolated	100.00 ± 0.00	98.08 ± 10.70	100.00 ± 0.00	0.25			
Item 24. Felt sad or low	95.65 ± 17.53	89.76 ± 26.68	99.13 ± 5.84	0.08			
Item 25. Worried about eyesight worsening	64.71 ± 33.83	41.51 ± 39.37	82.51 ± 22.33	**<0.01**	**0.01**	**0.03**	**<0.01**
Item 26. Worried about coping with everyday life	80.71 ± 28.62	70.51 ± 36.74	92.10 ± 20.46	**0.01**	0.40	0.20	**<0.01**
Item 27. Felt like a nuisance or a burden	98.55 ± 9.31	96.16 ± 14.88	99.13 ± 5.84	0.43			
Item 28. Interfered with life in general	85.62 ± 20.65	86.04 ± 21.09	92.58 ± 6.39	0.18			

IVI: Impact of Visual Impairment, SLE: Systemic Lupus Erythematosus. Statistical differences are highlighted in bold.

**Table 3 healthcare-12-00540-t003:** Automatic measurements using SS-OCT Triton.

	SLE Patients		Healthy Volunteers	*p*
	<10 Years	>10 Years		
Retinal thickness in ETDRS sector, µm				
Center	235.56 ± 31.88	242.33 ± 33.12	238.03 ± 30.02	0.70
Inner temporal	291.83 ± 22.68	297.83 ± 17.38	292.44 ± 26.72	0.54
Inner superior	305.75 ± 23.46	311.30 ± 16.04	309.76 ± 19.30	0.51
Inner nasal	305.15 ± 29.90	314.34 ± 18.37	309.85 ± 19.26	0.30
Inner inferior	304.06 ± 15.51	310.35 ± 18.48	309.00 ± 13.91	0.24
Outer temporal	251.85 ± 13.74	255.95 ± 12.07	250.85 ± 17.43	0.36
Outer superior	272.27 ± 13.78	274.77 ± 12.53	272.99 ± 16.62	0.79
Outer nasal	283.74 ± 20.56	291.43 ± 18.16	288.88 ± 15.82	0.23
Outer inferior	263.15 ± 14.68	268.25 ± 17.67	265.33 ± 17.10	0.48
Vascular density in superficial capillary plexus, %				
Center	21.77 ± 6.60	20.72 ± 6.02	22.24 ± 4.79	0.53
Temporal	45.91 ± 2.74	45.97 ± 2.95	45.40 ± 2.85	0.60
Superior	49.06 ± 1.88	49.64 ± 3.70	49.22 ± 3.10	0.69
Inferior	48.43 ± 3.54	48.02 ± 8.20	48.21 ± 4.45	0.95
Nasal	45.26 ± 3.01	44.31 ± 3.65	43.53 ± 4.96	0.14
Vascular density in deep capillary plexus, %				
Center	20.08 ± 7.21	20.30 ± 7.32	20.77 ± 8.41	0.91
Temporal	46.49 ± 2.45	47.07 ± 6.75	46.96 ± 3.50	0.85
Superior	50.36 ± 3.51	52.11 ± 3.97	51.32 ± 3.03	0.10
Inferior	52.64 ± 3.77	53.50 ± 5.00	53.70 ± 5.25	0.56
Nasal	46.34 ± 7.37	47.68 ± 3.92	48.17 ± 3.84	0.27

SLE: Systemic Lupus Erythematosus.

**Table 4 healthcare-12-00540-t004:** Clinical values in patients with SLE.

	SLE Patients		*p*
	<10 Years	>10 Years	
Duration of SLE, months	67.25 ± 38.76	164.70 ± 47.08	**<0.01**
Duration of treatment with HCQ, months	60.72 ± 34.92	89.47 ± 46.57	**<0.01**
Daily dose of HCQ, mg/kg	3.46 ± 1.38	3.34 ± 0.92	0.68
Dose of HCQ, mg	221.95 ± 75.50	209.68 ± 58.34	0.44
Cumulative dose of HCQ, mg	407.50 ± 270.58	560.50 ± 334.25	**0.04**
SLEDAI	1.68 ± 1.47	2.16 ± 1.79	0.22
SLICC	0.29 ± 0.56	1.32 ± 1.78	**<0.01**
LIT score	33.49 ± 20.74	35.98 ± 22.66	0.63
Hb, mg/dL	13.47 ± 1.17	13.09 ± 2.08	0.33
Hematocrit, %	39.90 ± 3.62	52.59 ± 71.18	0.26
Fe, µg/dL	85.80 ± 37.83	80.59 + 43.83	0.61
Ferritin, ng/mL	75.98 + 98.94	94.66 ± 76.33	0.43
Vitamin D, nmol/L	65.66 ± 31.05	63.78 ± 28.63	0.80
Ca, mg/dL	9.53 ± 0.61	9.49 ± 0.40	0.75

SLE: Systemic Lupus Erythematosus, HCQ: hydroxychloroquine, SLEDAI: Systemic Lupus Erythematosus Disease Activity Index, SLICC: Systemic Lupus International Collaborating Clinic index, LIT: Lupus Impact Tracker, Hb: hemoglobin. Statistical differences are highlighted in bold.

**Table 5 healthcare-12-00540-t005:** Factors influencing vision-related quality of life in linear regression analysis.

**SLE Patients < 10 Years**								
** Reading and Information** **(Adjusted R^2^ = 0.99)**			**Mobility and Independence** **(Adjusted R^2^ = 0.72)**			**Emotional Well-Being** **(Adjusted R^2^ = 1.00)**		
**Factor**	**β (*p*)**	**R^2^ Change**	**Factor**	**β (*p*)**	**R^2^ Change**	**Factor**	**β (*p*)**	**R^2^ Change**
Ferritin	−0.11 (<0.01)	0.37	Ferritin	−0.13 (0.01)	0.53	Ferritin	−0.27 (<0.01)	0.45
BCVA	−17.32 (<0.01)	0.23	LIT	−0.31 (0.01)	0.23	BCVA	−22.07 (<0.01)	0.27
Age	−1.00 (<0.01)	0.18	Constant	109.59 (<0.01)		ETDRS outer temporal	0.61 (<0.01)	0.12
Ca	20.51 (<0.01)	0.09				VD deep nasal	0.34 (<0.01)	0.09
Mean dose of HCQ	−0.56 (<0.01)	0.09				VD superficial superior	−3.07 (<0.01)	0.06
Vitamin D	0.08 (<0.01)	0.02				BMI	−0.28 (<0.01)	0.01
VD deep nasal	−0.17 (<0.01)	0.01				Vitamin D	0.03 (0.01)	0.01
Constant	50.06 (<0.01)					Constant	93.58 (<0.01)	
**SLE patients > 10 years**								
**Reading and informatio**n **(adjusted R^2^ = 0.89)**			** Mobility and independence** **(adjusted R^2^ = 1.00)**			** Emotional well-being** **(adjusted R^2^ = 1.00)**		
**Factor**	**β (*p*)**	**R^2^ Change**	**Factor**	**β (*p*)**	**R^2^ Change**	**Factor**	**β (*p*)**	**R^2^ Change**
BMI	−2.31 (<0.01)	0.59	Hb	7.55 (<0.01)	0.95	SLICC	−8.49 (<0.01)	0.77
Vitamin D	−0.43 (0.01)	0.33	SLICC	−8.02 (<0.01)	0.05	BMI	−1.32 (<0.01)	0.14
Constant	162.55 (<0.01)					HCQ duration	0.13 (<0.01)	0.08
						SLE duration	−0.07 (0.02)	0.01
						Constant	129.50 (<0.01)	

SLE: Systemic Lupus Erythematosus, BCVA: best-corrected visual acuity, VD: vascular density, LIT: Lupus Impact Tracker questionnaire, BMI: body-mass index, Hb: hemoglobin, SLICC: Systemic Lupus International Collaborating Clinic index, HCQ: hydroxychloroquine.

**Table 6 healthcare-12-00540-t006:** Bivariate correlations between IVI and LIT questionnaires.

		LIT Questionnaire	
		SLE Patients <10 Years	SLE Patients >10 Years
**IVI Questionnaire**	Reading and accessing information	0.60 (<0.01)	0.82 (<0.01)
	Mobility and independence	−0.34 (0.03)	−0.33 (<0.07)
	Emotional well-being	0.59 (<0.01)	0.69 (<0.01)

IVI: Impact of Vision Impairment, LIT: Lupus Impact Tracker, SLE: Systemic Lupus Erythematosus.

## Data Availability

The data are available at the Miguel Servet University Hospital in Zaragoza (Spain).

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
