# Peer review of "Vision-Related Quality of Life in Patients with Systemic Lupus Erythematosus"

_healthcare, 2024, doi:10.3390/healthcare12050540_

Round 1

Reviewer 1 Report

Comments and Suggestions for Authors

Here are my specific comments.

Line 72: How was SLE diagnosed in these participants? Did they come for the eye exam after being diagnosed?

Line  73: How long does it take for SLE to show Ophthalmological symptoms/signs? Why was 10 years chosen as a cut off?

Line 74: Why is axial length stated as a criteria and not refractive error?

Line 75: Looking at the exclusion criteria, I am wondering what were the presenting symptoms of these patients. Why did they come for an eye exam?

Line 82: a ?thorough ophthalmological exam. Spelling error

Line 87: what is the importance of conducting all the tests within a specific period of time for this study? It will benefit the readers if this is specified in the paper. In general, specifying the rationale behind each selection criteria and tests will benefit the readers to understand their importance pertaining to SLE and the aim of the study.

Line 92,93: The authors can consider changing the wording to something like … to fill out the questionnaire.

Line 92: Mention the reference for the IVI questionnaires where it is first introduced in the paper.

Line 96: What were the options for the questions that the subjects were to choose from? Was it Likert’s scale?

Line 100: The authors could expand a bit on LIT to say what were the categories of questions and how these categories were different from those of VRQoL.

Line 117: what is the prevalence of SLE in the region where the study was conducted?

Line 117: It will be interesting to know how many participants with SLE<10 years had eye disorders and had to be excluded because of the strict exclusion criteria.

Line 117: 72 participants in 2 years. That is a lot of patients with SLE. I am wondering whether the prevalence of SLE in the region where the study was conducted is larger or were the participants invited to participate in the study from the hospital’s medical records?

Table 2: What is item 15: getting information mean in the context of IVI?

Line 135: Any reason for why group 2 started their treatment late? Converting duration in years, the mean duration of SLE in group 1 was 6 years, and the treatment was for 5 years, on the other hand, group 2 had it for 14 years, and the treatment duration was 7 years. Could this have increased the disease impact in group 2?

Line 135: please change diary to daily.There are many spelling errors throughout the manuscript. Kindly address that.

Line 143: what are the statistical significance values for vitamin D and Ca

Lines 133-143: can be presented in a table for better access to information.

Table 5: why is mobility and independence negatively correlated with LIT?

Table 5: You can consider expanding other acronym LIT, IVI, to maintain consistency across tables.

Line 161: what are SLE strategies? Are you missing a word there?

Line 162: Did you mean that the HRQoL does not consider the impact of reduced vision on health? Please rephrase the sentence for better clarity.

Line 176: In item 25, the p value is < 0.05 for groups 1 and 2.

Line 188: which parameter are you referring to by saying systemic disease activity scales? Use the same words or acronym to refer to something to avoid confusion.

Line 183: Any reason why serum ferritin was not a significant factor for SLE > 10 years?

Line 197: low serum levels or low serum vit D levels?

Line 205: Dry eye can be better evaluated using other sensitive tests than Schirmer test, perhaps there was some reflexive tearing.

Line 201: why did the authors think AL and IOP should have influenced VRQoL?

Line 212: Were any questions in VRQoL overlapped with questions in LIT?

Comments on the Quality of English Language

There are a few spelling errors and some of the sentences could be rephrased for better understanding.

Author Response

Dear reviewer #1,

Thank you very much for your highly valuable recommendations. We really appreciate your time revising our manuscript. As suggested, the following changes have been made to the manuscript:

  • Line 72: How was SLE diagnosed in these participants? Did they come for the eye exam after being diagnosed?
    1. It has been explained. EULAR/ACR criteria were used by the department of internal medicine.
  • Line  73: How long does it take for SLE to show Ophthalmological symptoms/signs? Why was 10 years chosen as a cut off?
    1. SLE might never show ophthalmological symptoms or signs. It is expected that patients with a longer course of the disease, should have higher probabilities of ophthalmological affection. However, there is no clinical evidence, nor any clinical guidelines, establishing an exact risk time for this complication.
  • Line 74: Why is axial length stated as a criteria and not refractive error?
    1. Because refractive error may be due to a high corneal astigmatism, or due to a flat cornea. Curvature of anterior or posterior cornea has never been related to retinal disorders. In contrast, axial length has been related with retinal disorders. In fact, most clinical studies prefer to use axial length values instead of refractive error. An example is high myopia, in which higher values of axial length are related to posterior staphylomas, higher degrees of chorio-retinal atrophy, and neovascular membrane. Axial length was decided to be used in this study because it is related to retinal affection, and not to corneal parameters or aberrometric outcomes.
  • Line 75: Looking at the exclusion criteria, I am wondering what were the presenting symptoms of these patients. Why did they come for an eye exam?
    1. No symptoms were regarded as exclusion criteria. ‘Exclusion criteria comprised: any ophthalmological disorder, previous ophthalmological treatment, retinopathy secondary to HCQ, ophthalmological disorders secondary to SLE, any other disease different from SLE or Sjögren syndrome, or treatment with corticosteroids with a dose higher than 5 mg/d’
    2. They came for the eye exam because of the study. ‘Patients signed written informed consent after being informed about the study.’
  • Line 82: a ?thorough ophthalmological exam. Spelling error
    1. The spelling error has been corrected.
  • Line 87: what is the importance of conducting all the tests within a specific period of time for this study? It will benefit the readers if this is specified in the paper. In general, specifying the rationale behind each selection criteria and tests will benefit the readers to understand their importance pertaining to SLE and the aim of the study.
    1. The period of time is provided for a better reliability, although there is still no evidence that it might be related to different outcomes. In case you still find it out of place, we will delete it from the manuscript.
  • Line 92,93: The authors can consider changing the wording to something like … to fill out the questionnaire.
    1. It has been changed
  • Line 92: Mention the reference for the IVI questionnaires where it is first introduced in the paper.
    1. It has been done. Now it is in the introduction section.
  • Line 96: What were the options for the questions that the subjects were to choose from? Was it Likert’s scale?
    1. Possible answers to the questionnaire have been added.
  • Line 100: The authors could expand a bit on LIT to say what were the categories of questions and how these categories were different from those of VRQoL.
    1. There are no categories in LIT questionnaire. As we say in the manuscript, ‘this questionnaire has no vision specific items’. Therefore, it is completely different from IVI questionnaire.
  • Line 117: what is the prevalence of SLE in the region where the study was conducted?
    1. It has been added in the second line, in the introduction.
  • Line 117: It will be interesting to know how many participants with SLE<10 years had eye disorders and had to be excluded because of the strict exclusion criteria.
    1. It has been added in the first line in the results section.
  • Line 117: 72 participants in 2 years. That is a lot of patients with SLE. I am wondering whether the prevalence of SLE in the region where the study was conducted is larger or were the participants invited to participate in the study from the hospital’s medical records?
    1. Epidemiology of SLE in the region has been added, as answered in a previous issue.
    2. The way patients were recruited has been added at the end of the first paragraph in Material and Methods section.
  • Table 2: What is item 15: getting information mean in the context of IVI?
    1. Some words were missing. It is: ‘getting information that you need’, as provided by the creator of this questionnaire.
  • Line 135: Any reason for why group 2 started their treatment late? Converting duration in years, the mean duration of SLE in group 1 was 6 years, and the treatment was for 5 years, on the other hand, group 2 had it for 14 years, and the treatment duration was 7 years. Could this have increased the disease impact in group 2?
    1. A paragraph has been added at the end of the discussion section.
  • Line 135: please change diary to daily. There are many spelling errors throughout the manuscript. Kindly address that.
    1. It has been changed.
  • Line 143: what are the statistical significance values for vitamin D and Ca
    1. P values have been added.
  • Lines 133-143: can be presented in a table for better access to information.
    1. They have been converted to a table.
  • Table 5: why is mobility and independence negatively correlated with LIT?
    1. Higher LIT scores mean higher affection in daily life. Lower mobility and independence mean a higher affection in daily life.
  • Table 5: You can consider expanding other acronym LIT, IVI, to maintain consistency across tables.
    1. They have been explained.
  • Line 161: what are SLE strategies? Are you missing a word there?
    1. It has been changed.
  • Line 162: Did you mean that the HRQoL does not consider the impact of reduced vision on health? Please rephrase the sentence for better clarity.
    1. It has been rephrased.
  • Line 176: In item 25, the p value is < 0.05 for groups 1 and 2.
    1. It has been corrected.
  • Line 188: which parameter are you referring to by saying systemic disease activity scales? Use the same words or acronym to refer to something to avoid confusion.
    1. It has been explained.
  • Line 183: Any reason why serum ferritin was not a significant factor for SLE > 10 years?
    1. No reason was found after the deep review of the presented outcomes.
  • Line 197: low serum levels or low serum vit D levels?
    1. It has been changed. ‘Lower serum vitamin D levels’
  • Line 205: Dry eye can be better evaluated using other sensitive tests than Schirmer test, perhaps there was some reflexive tearing.
    1. Schrimer test is one of the most objective tests for the diagnosis of the dry eye syndrome. Symptoms and signs are poorly correlated in this syndrome. There are two forms of Schirmer test. Type I is performed with no previous anesthesia, and it measures basal and reflex tearing. Type II is performed after topical anesthesia, and it only measures basal tearing. Those patients with severe affection in Schirmer I test, are expected to present more aggressive ophthalmological consequences. SLE is related to aqueous-deficient dry eye. This means, that both basal and reflex tearing are affected.
  • Line 201: why did the authors think AL and IOP should have influenced VRQoL?
    1. We did not think they had any influence on VRQoL. The statistical analysis showed that these variables had no influence on VRQoL.
  • Line 212: Were any questions in VRQoL overlapped with questions in LIT?
    1. Both questionnaires were completely different. LIT questionnaire has no vision-related items. All the items in IVI questionnaire are vision-related.

Reviewer 2 Report

Comments and Suggestions for Authors

I carefully read the manuscript and found it suitable for publication in the journal. I accept this article for possible publication.

The principal objective was analyzed the impact of SLE in vision-related quality of life, the subjects enrolled was divided in two groups: with less of ten years or more of ten years with SLE. All comparations was realized with age-matched healthy volunteers. At all participants, was performed complete slit-lamp examination and OCT. The IVI questionnaire applied has been validated in different ocular conditions and different levels of visual ability, and LIT questionnaire also validated and used by monitored the impact os SLE on lives of patents.

In general, the study was good designed and the conclusions are supported by the results. I have a few minor comments:

 1.    Introduction can be improved; I suggest include the purpose of questionnaires as specific instruments to measure the life quality.

2.    In the Results, include a short description of table 2, 3 and 4 for example indicate was statistical differences or not between groups.

3.    In the page 5 lines 133 to 143, the authors present the description of clinical records and clinical analysis but not are includes/represent in some table.

Author Response

Dear reviewer #2,

Thank you very much for your highly valuable recommendations. We really appreciate your time revising our manuscript. As suggested, the following changes have been made to the manuscript:

  • Introduction can be improved; I suggest include the purpose of questionnaires as specific instruments to measure the life quality.
    1. It has been improved with different vision-related quality of life questionnaires.
  • In the Results, include a short description of table 2, 3 and 4 for example indicate was statistical differences or not between groups.
    1. A short description has been added for the three tables.
  • In the page 5 lines 133 to 143, the authors present the description of clinical records and clinical analysis but not are includes/represent in some table.
    1. They have been converted to a table.

Reviewer 3 Report

Comments and Suggestions for Authors

I think this paper is a valuable research result that investigated the visual quality of SLE patients taking HCQ. However, I have some questions, which I will list below.

â‘     Abstract Line32, Line33.

The problem is that the abbreviations IVI and HRQoL are suddenly listed here.

â‘¡    3page, Line117.

The target eyes were usually both eyes when the ophthalmological examination was performed, and I believe that the answers to the IVI questionnaire are the results of observations made with both eyes, but these points are ambiguous.

â‘¢    2page, Line75.

It states that cases with retinopathy due to HCQ were excluded. However, I think it is inadequate that the diagnostic criteria for retinopathy caused by HCQ are not listed.

â‘£    6page, Line176

The biggest question in this paper is this phrase. This is because we can exclude the presence of HCQ retinopathy based on the results of visual acuity tests, ophthalmoscopy tests, and CRT-only tests in a subject group who have been taking HCQ for an average of 5 to 7 years. Is it possible?

I thought that fundus autofluorescence, color vision testing, and visual field testing were essential to deny the presence of HCQ retinopathy.

Based on the above points, I believe that the author should revise the idea on page 7, line 222, by going into more detail.

That's all from me.

Author Response

Dear reviewer #3,

Thank you very much for your highly valuable recommendations. We really appreciate your time revising our manuscript. As suggested, the following changes have been made to the manuscript:

  • Abstract Line32, Line33. The problem is that the abbreviations IVI and HRQoL are suddenly listed here.
    1. They are explained now.
  • 3page, Line117. The target eyes were usually both eyes when the ophthalmological examination was performed, and I believe that the answers to the IVI questionnaire are the results of observations made with both eyes, but these points are ambiguous.
    1. You are right. It has been explained in methods section. No vision-related questionnaire is capable of distinguishing whether the affection of the quality of life is due to one or the other eye. Items in IVI questionnaire appear in table 2.
  • 2page, Line75. It states that cases with retinopathy due to HCQ were excluded. However, I think it is inadequate that the diagnostic criteria for retinopathy caused by HCQ are not listed.
    1. They have been listed in the third paragraph in Material and methods section. All the tests recommended by the American Academy of Ophthalmology were performed.
  • 6page, Line176. The biggest question in this paper is this phrase. This is because we can exclude the presence of HCQ retinopathy based on the results of visual acuity tests, ophthalmoscopy tests, and CRT-only tests in a subject group who have been taking HCQ for an average of 5 to 7 years. Is it possible?
    1. We performed all tests according to the American Academy of Ophthalmology. We did not include any patient in this study with HCQ because it was an exclusion criterion. We did not include data about visual field, fundus autofluorescence or electroretinogram because the purpose of our study was to assess vision-related quality of life in patients with systemic lupus erythematosus under treatment with hydroxychloroquine, and to find the influencing factors.
    2. https://www.aao.org/education/clinical-statement/revised-recommendations-on-screening-chloroquine-h

Round 2

Reviewer 3 Report

Comments and Suggestions for Authors

None